# Circulating MicroRNAs as a Tool for Diagnosis of Liver Disease Progression in People Living with HIV-1

**DOI:** 10.3390/v14061118

**Published:** 2022-05-24

**Authors:** Miguel Angel Martinez, Cristina Tural, Sandra Franco

**Affiliations:** 1IrsiCaixa, Hospital Universitari Germans Trias i Pujol, Universitat Autònoma de Barcelona (UAB), 08916 Badalona, Spain; sfranco@irsicaixa.es; 2Internal Medicine Department, Hospital Universitari Germans Trias i Pujol, Universitat Autònoma de Barcelona (UAB), 08916 Badalona, Spain; ctural.germanstrias@gencat.cat

**Keywords:** miRNAs, liver injury, biomarker, diagnosis, prognosis, HIV-1

## Abstract

MicroRNAs (miRNAs) are small, non-coding RNAs that post-transcriptionally regulate gene expression by binding specific cell mRNA targets, preventing their translation. miRNAs are implicated in the regulation of important physiological and pathological pathways. Liver disease, including injury, fibrosis, metabolism dysregulation, and tumor development disrupts liver-associated miRNAs. In addition to their effect in the originating tissue, miRNAs can also circulate in body fluids. miRNA release is an important form of intercellular communication that plays a role in the physiological and pathological processes underlying multiple diseases. Circulating plasma levels of miRNAs have been identified as potential disease biomarkers. One of the main challenges clinics face is the lack of available noninvasive biomarkers for diagnosing and predicting the different stages of liver disease (e.g., nonalcoholic fatty liver disease and nonalcoholic steatohepatitis), particularly among individuals infected with human immunodeficiency virus type 1 (HIV-1). Liver disease is a leading cause of death unrelated to acquired immunodeficiency syndrome (AIDS) among people living with HIV-1 (PLWH). Here, we review and discuss the utility of circulating miRNAs as biomarkers for early diagnosis, prognosis, and assessment of liver disease in PLWH. Remarkably, the identification of dysregulated miRNA expression may also identify targets for new therapeutics.

## 1. Introduction

MicroRNAs (miRNAs) are single-stranded 19 to 22-nucleotide noncoding RNAs. In animals, miRNAs act as post-transcriptional inhibitors of gene expression. In association with proteins of the Argonaute family, miRNAs form an RNA-induced silencing complex (RISC) that targets specific mRNA transcripts [1]. miRNAs specifically bind target mRNAs, usually in the 3′ untranslated region (UTR), resulting in a reduction in mRNA translation (Figure 1A). The net result is reduced protein production from the targeted mRNA. The control of gene expression allows miRNAs to regulate an important number of cell processes and all of an organism’s physiology and metabolism. Indeed, miRNAs are key regulators of biological processes in animals.

Massive techniques for small RNA sequencing have allowed the identification and annotation of nearly 2000 human miRNAs [2]. Because other small non-coding RNAs, such as transfer RNAs (tRNAs), small nuclear RNAs (snRNAs), small nucleolar RNAs (snoRNA), small interfering RNAs (siRNAs), or Piwi-interacting RNAs (piRNAs), can be wrongly annotated as miRNAs, the number of human miRNAs may have been overestimated [3]. Applying unique and mechanistically well understood features of the different miRNAs has allowed the identification of 556 bona fide human miRNAs [3]. Based on their sequences, these human miRNAs can be grouped into different miRNA families, some of which are well conserved across the animal kingdom [4]. The conservation of miRNA through evolution reveals the important biological functions of conserved mammalian miRNAs.

MicroRNAs are implicated in almost every cellular process and, consequently, are critical for cell differentiation, homeostasis, and animal development [5]. Computational analyses indicate that more than 50% of the human transcriptome may be regulated by at least one miRNA [6]. Deletion of essential factors in miRNA biogenesis (e.g., Dicer) is lethal for animal development [7]. Similarly, deregulation of miRNA activity is associated with several animal and human diseases (Figure 1B). Mechanisms that govern miRNA function include miRNA stability, sequence editing, transport from the cytoplasm, target interactions, viral factors, and post-translational modifications of Argonaute proteins [5]. Since the discovery of miRNA in the early 1990s, the alteration in cellular miRNA levels was observed to potentially be a sentinel marker of something going wrong in a particular cell or tissue and opened up new avenues for diagnosis. Anomalous expression of miRNAs has been found to be associated with multiple human diseases including cancers, viral infections, liver injury, diabetes, cardiovascular and neurodegenerative diseases, as well as other disorders. Similarly, it was evident that approaches to modulate cellular miRNA levels offered new targets for therapeutic interventions [8].

Initially, miRNAs were suggested to regulate the expression of their target mRNAs only in cells in which they were produced. Later, miRNAs were found to be secreted into extracellular fluids and delivered to other cells in which they can also function [9]. As miRNAs can be secreted into body fluids, and then be taken up by cells in other tissues, they can be called RNA hormones. The role of miRNAs as signaling and coordination molecules between cells and organs has positioned them as excellent disease biomarkers [10,11,12,13]. In addition, miRNAs are easily accessible in body fluids, such as blood, urine, or saliva, using noninvasive methods, and they are stable and readily measurable by several techniques.

Obesity is spreading worldwide, with an increasing prevalence of insulin resistance, diabetes, and chronic liver disease, making it a major public health burden [14,15]. Liver disease is most commonly caused by chronic hepatitis B and C, alcohol-associated liver disease (ALD), and nonalcoholic fatty liver disease (NAFLD). Liver disease can further progress to inflammation (nonalcoholic steatohepatitis (NASH)) and fibrosis, and finally to cirrhosis and hepatocellular carcinoma (HCC) as end-stage diseases (Figure 2) [16]. Two million deaths worldwide are caused by cirrhosis, viral hepatitis, and HCC each year [14,15]. Alterations in non-coding RNA-dependent regulation, particularly miRNAs, play an important role in hepatocyte function, liver injury, viral hepatitis, ALD, NAFLD, liver fibrosis progression, and HCC [17,18]. Despite the complexity of miRNA biology and liver metabolism, miRNAs are emerging as reliable circulating biomarkers for the non-invasive diagnosis of different stages of liver disease [16,18].

After the success of highly active antiretroviral therapy (ART), chronic, noninfectious comorbidities are increasingly important in caring for the aging population of people living with human immunodeficiency virus type 1 (PLWH). Liver disease is a leading cause of death unrelated to acquired immunodeficiency syndrome (AIDS) among PLWH [20]. Human immunodeficiency virus type 1 (HIV-1) infections may cause liver disease through several mechanisms. Liver stellate and Kupffer cells can be directly infected by HIV-1. Moreover, HIV-1 can cause chronic inflammation, microbial product translocation, and low-grade disseminated coagulopathy [21]. Thus, progression to liver fibrosis, cirrhosis, and HCC are often complications in PLWH. Liver disease can be further exacerbated by coinfection with chronic hepatitis C virus (HCV) and hepatitis B virus (HBV) and/or drug-induced hepatotoxicity (Figure 2) [22,23]. Abnormal liver enzymes are common in HIV-1-infected individuals, even in the absence of viral hepatitis or alcohol abuse [24]. Similarly, there are an increasing number of reports on the development of NAFLD or NASH in PLWH. Bacterial translocations [25] and heart failure [26] have also been associated with HIV-1-associated liver disease progression. This scenario emphasizes the necessity of new biomarkers of liver disease progression in PLWH.

There is a lack of noninvasive biomarkers to identify individuals that will progress to liver fibrosis and decompensated cirrhosis and are more prone to developing complications. In this review, we discuss the utility of circulating miRNAs to identify HIV-1-infected individuals at risk of developing end-stage liver disease.

## 2. HIV-1 and Liver Disease

The introduction of ART in the mid-1990s drastically reduced the number of AIDS-related and non-AIDS-related deaths among HIV-1-infected individuals [27]. Nevertheless, liver disease remains one of the major non-AIDS-related causes of morbidity and mortality amongst PLWH [19,20]. Compared to uninfected individuals, a higher frequency (range 11–40.9%) of liver fibrosis has been found in PLWH without coinfection with hepatitis viruses or alcohol abuse [28,29]. This finding strongly suggests that HIV-1 infection itself contributes to liver disease progression. A reduction in the number of plasma CD4+ T cells correlates with rapid liver fibrosis progression in HIV-1-monoinfected individuals. Gut CD4+ T-cell depletion increases gut permeability and microbial translocation, and the associated liver macrophage Kupffer cell response generates an environment that drives liver inflammation and fibrosis [28] (Figure 2).

Another factor affecting liver disease progression in PLWH is ART hepatotoxicity [30], which can be clinically relevant to long-term treatment. Although the association of liver fibrosis and ART has been controversial over the years, first-generation nucleoside reverse transcriptase inhibitors, particularly didanosine and zidovudine, appear to carry the greatest risk for the progression of liver fibrosis (Figure 2). In contrast, integrase and entry inhibitors seem to have the lowest risk. Remarkably, protease inhibitors, which have been associated with insulin resistance and obesity [31], do not show a significant risk for liver fibrosis [30].

Similar to an HIV-1-uninfected population, NAFLD, defined as hepatic steatosis without significant alcohol consumption or coinfection with hepatic viruses, is a significant cause of liver disease in PLWH [32]. NAFLD can further progress to NASH, fibrosis, cirrhosis, and HCC [32,33,34]. NAFLD is also associated with cardiovascular disease, which is currently one the main problems in the long-term management of PLWH [35]. Risk factors for NAFLD in the general population are those associated with metabolic syndrome, including obesity, type 2 diabetes mellitus and associated insulin resistance, dyslipidemia, and hypertension [32,33]. Importantly, the risk of developing metabolic syndrome is double in PLWH [32]. Although the idea that PLWH progress faster to NASH is controversial, the prevalence of NASH and liver fibrosis in PLWH may be double that of HIV-1-negative individuals [36,37]. Microbial translocation, direct impact of HIV-1 infection, or ART exposure have been associated with the high prevalence of NAFLD and NASH observed in PLWH [33]. Host genetic factors that increase the prevalence of NAFLD in the general population, such as polymorphisms in the genes encoding patatin-like-phospholipase domain-containing 3 (PNPLA3) and transmembrane 6 superfamily member 2 (TM6SF2), may also impact liver disease in PLWH [38,39]. In addition to NAFLD, alcohol consumption is also associated with faster liver fibrosis progression in PLWH patients who are not coinfected with hepatitis viruses [40]. Non-invasive and early detection of NAFLD and its progression to NASH to identify those at highest risk of advanced liver disease is still challenging and greatly needed in the clinic.

HBV and HCV cause chronic hepatic infection and disease and constitute two of the main sources of liver disease, cirrhosis, and HCC, infecting 300 million and 70 million people worldwide, respectively [41,42]. HBV and HCV share transmission routes with HIV-1 and, thus, the prevalence of coinfection with the two viruses is high in PLWH. HBV and HCV coinfection rates in PLWH are approximately 10% and 6%, respectively, depending of the endemicity area [43,44]. HCV/HIV-coinfection accelerates liver disease progression to liver cirrhosis and HCC, even in the presence of ART [45]. HIV-1 coinfection directly accelerates HCV-mediated liver fibrosis by increasing hepatocyte apoptosis, oxidative stress, and fibrogenesis and increasing hepatic stellate and Kupffer cells inflammation, activation, and fibrogenesis. Indirectly, HIV-1 also increases liver disease by increasing microbial translocation, impairing HCV immune responses, and augmenting HCV replication [45]. HCV may also impact the natural history of HIV-1 by impairing immune responses to HIV-1 and accelerating the progression to AIDS [46]. Remarkably, HIV-1 infection increases metabolic abnormalities, such as insulin resistance, that result in hepatic steatosis and triglyceride accumulation in hepatocytes, which contribute to accelerating HCV-associated end-stage liver disease [47]. HCV/HIV-coinfected subjects have a higher mortality risk and more severe HCC than HCV-monoinfected individuals, indicating the existence of pathways unique to PLWH that contribute to accelerating liver disease. Similarly, HBV/HIV-1-coinfection alters the natural history of HBV and accelerates the progression of chronic HBV to cirrhosis, end-stage liver disease, or HCC compared to chronic HBV monoinfection [48]. HIV-1-induced depletion of CD4+ T cells seems to enhance HBV replication and augment liver injury. [49]. In addition, HBV/HIV-1 coinfection induces hepatocyte fibrogenesis, increases hepatocyte apoptosis, and changes the hepatic cytokine environment (e.g., elevated C-X-C motif chemokine 10) [50]. As mentioned with HCV/HIV-1 coinfection, gut depletion of CD4+ T cells may increase microbial translocation and concomitant liver inflammation.

Importantly, HCV and HBV infections have become clinically manageable diseases. Thanks to highly effective treatments (i.e., direct-acting antivirals (DAAs)), HCV is now a curable disease in nearly 95% of coinfected individuals. Thus, HCV eradication is an attainable objective, at least in resource-rich settings. Similarly, chronic hepatitis B is now controlled by long-term treatment with HBV polymerase inhibitors, such as tenofovir. Therefore, obesity, metabolic syndrome, and the resulting NAFLD and NASH have gained clinical importance in PLWH.

## 3. MicroRNAs and Liver Disease

Dysregulation of miRNA expression (Figure 1B) and activity is highly associated with metabolic disorders that promote NAFLD and progression to liver disease [16]. Several miRNAs are implicated in diverse aspects of liver lipid metabolism. In particular, miRNA-122, miRNA-34a, miRNA-33, miRNA-21, miRNA-375, miRNA-192, miRNA-100, miRNA-155, miRNA29a, and miRNA-223 perform relevant regulatory functions in liver metabolism (reviewed in [16,51]; Table 1). For example, a recent study found that circulating levels of miRNA-34a, miRNA-122, and miRNA-192 are independently associated with hepatic steatosis and fibrosis [52]. miRNA-122 represents 70% of all miRNA copies found in the liver [53] and is a major regulator of hepatic lipid metabolism. miR-122 expression is downregulated in the livers of individuals with hepatic steatosis. Reducing the liver levels of miRNA-122 downregulates lipogenic enzymes, increases fatty acid β-oxidation, and decreases accumulation of intracellular triglycerides and cholesterol synthesis [54]. miR-122 has also been shown to be required for the replication of HCV [55]. miRNA-34a, though weakly expressed in hepatocytes, is upregulated in individuals with NASH [56]. miRNA-34a restrains fatty acid catabolism, favors steatosis development, antagonizes lipogenesis by targeting AMP-activated protein kinase α activity, and controls lipid storage by specifically targeting hepatocyte nuclear factor 4 [57]. Circulating levels of miRNA-33 are also upregulated in individuals with NAFLD or NASH [58]. This miRNA is associated with obesity and regulates both cholesterol and fatty acid metabolism by targeting sterol regulatory element-binding proteins and ATP binding cassette subfamily A member 1 [59]. miRNA-21 is strongly upregulated in individuals with NASH [60]. miR-21 deletion in mice limits steatosis development induced by an obesogenic diet through upregulation of multiple miRNA-21 targets involved in lipid metabolism [61]. miRNA-375, which is also elevated in livers with steatosis [62], is highly expressed in the β cells of the pancreatic islets and is involved in insulin secretion and glucose homeostasis [63]. Deletion of miRNA-375 induces a marked decrease in β cell number, provoking a severe diabetic state. miRNA-192 is deregulated in the livers of individuals with liver disease [62]. miRNA-192 is related to lipid synthesis by targeting stearoyl-CoA desaturase 1 [64]. Intriguingly, it is upregulated in the livers of individuals with NAFLD and downregulated in individuals with NASH [16]. Elevated levels of mRNA-100 are also observed in individuals with NAFLD [65]. mRNA-100 targets repressors of lipogenic transcription factors, such as the nuclear receptor corepressor [66]. miRNA-155 is a multi-functional miRNA known to regulate numerous biological processes, including cholesterol and fatty acid metabolism pathways in the liver. miRNA-155 deregulation has been implicated in the pathogenesis of fatty liver disease [67]. miRNA-155 controls functional networks in the liver related to injury responses, steatosis, inflammation, fibrosis, and carcinogenesis [68]. miR-29 is directly involved in the regulation of lipid metabolism by downregulating hepatic lipogenesis [69]. miR-29a serum levels are significantly down-regulated in individuals with NAFLD [70]. Finally, miRNA-223 is a critical regulator of NASH progression by targeting several inflammatory genes and oncogenes [71]. The highest levels of miRNA-223 are expressed in neutrophils and it plays a critical role in attenuating neutrophil maturation and activation [72]. miRNA-223 performs an important role in attenuating ALD-induced liver injury by targeting interleukin-6 and inhibiting the nuclear factor kappa B kinase subunit alpha genes in neutrophils [71]. The miRNAs described here, as well as many others, may also indirectly affect liver lipid metabolism by targeting carbohydrate metabolism, stress-activated pathways, and tumor suppressors. The miRNAs described here are not the sole miRNAs implicated in liver metabolism, but they have been described as the most relevant therapeutic targets or biomarkers of human liver disease [16,51].

Unravelling the pathophysiological functions of miRNAs has promoted their use as potential therapeutic targets [8]. Given the potential of miRNAs to simultaneously repress multiple lipid or carbohydrate metabolism pathways or tumor or inflammation processes, one potential therapeutic strategy is to introduce a miRNA mimic or inhibitor to restore normal cell and organ functionality. A phase II clinical trial with miRNA-122 antisense oligonucleotide demonstrated that it is possible to reduce viral RNA loads in individuals with HCV infection without inducing viral resistance [73]. An antisense oligonucleotide that sequesters a mature miRNA in a highly stable heteroduplex is able to strongly inhibit its function. In addition, administration of a mature miRNA, such as miRNA-34a, results in antitumor activity in a subset of individuals with refractory advanced solid tumors [74]. miRNA-34a is lost or expressed at reduced levels in association with a loss of p53 function, and its de novo delivery reduces tumor cell growth. In cultured hepatocytes, overexpression of miR-223 ameliorates NAFLD and HCC by targeting multiple inflammatory and oncogenic genes [71]. Another example is miRNA-194; overexpression of miRNA-194 in hepatoma cells increases their sensitivity to treatment for HCC, such as sorafenib, by targeting Ras-related C3 botulinum toxin substrate 1 [75]. However, two major issues are obstacles for miRNA-based treatments [16]: better vehicles for specific miRNA delivery are needed, and undesired off-target effects. Inhibition of miRNA-21 may improve NAFLD/NASH progression [61], but may also induce a deficient immune response [76].

In addition to the possible use of miRNAs, or antisense inhibitors, in treatment, their main therapeutic application has been as disease biomarkers. miRNAs have been shown to have potential in detecting a wide variety of human pathologies. miRNAs are easily accessible in biofluids, such as blood, urine, breast milk, and saliva, through noninvasive means. Furthermore, they exhibit high stability and can be readily assessed by various methods (e.g., next generation sequencing, quantitative PCR) with high specificity. Circulating serum or plasma miRNA signatures have been widely studied for their biomarker potential. In particular, circulating miRNAs have been extensively searched as biomarkers of cancer diagnosis and prediagnosis [77]. Circulating miRNAs have been investigated to augment low-dose computed tomography for the diagnosis of lung cancer [78]. miRNAs have been suggested to be found in both serum and plasma as cell-free circulating RNAs, which may originate from dying cells or be actively secreted by cells under specific physiological circumstances. miRNAs can also be selectively secreted by exosomes [5]. Because exosomes contain selective miRNAs that differ from the cell or tissue of origin, secreted vesicles play an important role in normal physiological processes. Therefore, circulating miRNA signatures may signal physiological abnormalities. The relevance of miRNAs in the physiology and metabolism of the liver makes them ideal clinical markers for monitoring liver injury and associated metabolic disorders, such as NAFLD, as well as for predicting and diagnosing progression to NASH, cirrhosis, and HCC [79].

## 4. MicroRNAs as Biomarkers of Liver Disease Progression in HIV-1-Infected Patients

HIV-1 infection modifies the miRNA profile in plasma [65,80], peripheral blood mononuclear cells [81], and gastrointestinal mucosa [82]. These altered miRNA profiles can potentially serve as efficient biomarkers for HIV-1 diagnosis and disease progression [83]. The plasma miRNA signature can also predict the immune response after ART in PLWH [84]. Conversely, HIV-1 replication may be restricted by certain host cellular miRNAs, suggesting that miRNAs play a role in host defense and in maintaining latency within resting CD4+ T cells [85]. Interestingly, the differential expression of important protective histocompatibility locus antigen (HLA) alleles in HIV-1 infection has been shown to be regulated by miRNAs [86]. Overall, studies have emphasized the interplay of miRNAs in the biological cycle of HIV-1.

As suggested in previous sections, HIV-1 monoinfection-induced inflammation and/or long-term ART toxicity may contribute to the evolution of liver disease. Serum levels of miRNA-200a have been correlated with transaminase levels and the presence of liver steatosis in PLWH, suggesting that miRNA-200a may be a predictor of steatosis progression and probably liver cell injury [87]. However, this study only measured the levels of this miRNA. Recently, small RNA sequence analysis of plasma circulating miRNA in HIV-1 monoinfections and HCV monoinfections identified deregulated miRNAs previously associated with liver injury [65]. In this study, 14 miRNAs that were highly upregulated in individuals with HCV monoinfection were also upregulated in individuals with HIV-1 monoinfection that exhibited elevated transaminase levels or focal nodular hyperplasia. Specifically, circulating levels of miR-193b-5p, miR-125b-1-3p, miR-100-5p, miR-192-5p, miR-99a-5p, and miR-122-3p were highly correlated with transaminase levels and liver fibrosis stage in individuals monoinfected with HIV-1 [65] (Table 2, and Figure 3). Remarkably, miRNA-122-3p and miRNA-193b-5p were highly upregulated in HIV-1-monoinfected individuals with elevated transaminase levels or focal nodular hyperplasia, but not in HIV-1 individuals with normal levels of transaminases. These results indicate that HIV-1 infection impacts liver-related miRNA levels in the absence of an HCV coinfection, and show the potential of miRNAs as biomarkers for the progression of liver disease in PLWH.

Serum levels of miRNA-193, miRNA-125b, miRNA-100, miRNA-192, miRNA-99a, and miRNA-122 have been found to be deregulated in patients with HBV-associated HCC [89,90,91,92,93], confirming the implication of these miRNAs in liver metabolism. Depending on the method used to quantify the levels of circulating miRNAs, these miRNAs and others (e.g., miR-let-7b, miRNA-155, and miR-15b) have been found to be deregulated in individuals monoinfected with HCV or HBV [65,93,94,95,96]. HCV coinfection has been the leading cause of liver disease in PLWH. Currently, at least in developed countries, the introduction of HCV DAAs has almost eradicated HCV coinfections in PLWH. Nevertheless, HIV-1/HCV coinfections have provided a useful model for learning about liver disease in both HIV-1-monoinfected individuals and the general population.

A seminal study performed on 335 PLWH showed that circulating levels of miRNA-122, miRNA-22, and miRNA-34a were highly upregulated in HIV-1/HCV-coinfected individuals [97]. These miRNAs correlated with liver function tests and were independent predictors of liver injury. Importantly, miRNA-122 and miRNA-34a levels were significantly upregulated in HIV-1/HCV-coinfected individuals, miRNA-22 levels were highest in HIV-1/HBV-coinfections, and circulating levels of miRNA-34a correlated positively with illicit drug use and ethanol consumption. Intriguingly, liver biopsy analysis of HIV-1/HCV coinfection samples showed a significant downregulation of miRNA-122 and miRNA-192 [98]. Circulating levels of miRNAs have been suggested to not always reflect their tissue expression and/or activity [16]. More recently, large-scale deep sequencing analysis of the expression of small RNAs in plasma samples from 46 HIV-1/HCV-coinfected individuals that did not exhibit liver fibrosis at the time of sampling identified seven miRNAs (miRNA-100–5p, miRNA-192–5p, miRNA-99a-5p, miRNA-122–5p, miRNA-125b-2–3p, miRNA-1246, and miRNA-194–5p) that highly correlated with progression to liver fibrosis [88] (Table 2 and Figure 3). Two miRNAs, miRNA-100–5p and miRNA-192–5p, yielded a sensitivity of 88% and specificity of 85% for detecting liver fibrosis progression. These findings demonstrate that circulating miRNA levels have potential in predicting liver fibrosis progression before the clinical detection of liver fibrosis or significant clinical signs.

New DAA therapy efficiently eradicates HCV [99]. Elimination of HCV does not always result in the curing of liver disease, especially in patients with advanced fibrosis or cirrhosis, and particularly in PLWH coinfected with HCV. Fibrogenic signals persist in DAA-treated individuals after achieving HCV eradication [100], suggesting that individuals with HCV infection who eradicate the virus on DAA therapy still need to be monitored for signs of liver disease. In this scenario, quantification of the circulating levels of miRNAs may emerge as a noninvasive prognostic tool to assess long-term post-DAA clinical outcomes. Certainly, levels of circulating miRNA-122 in serum are significantly reduced in HCV-monoinfected individuals who eradicated the virus after treatment with DAAs [101]. In contrast, in individuals who do not eradicate the virus, miR-122 levels begin to return to baseline levels after the second week of treatment. Similarly, in contrast to non-responders, individuals who respond to antiviral treatment do not show augmented miR-155 in peripheral monocytes [94]. A study that analyzed the evolution of the miRNA signature in circulating peripheral blood mononuclear cells from HIV-1/HCV-coinfected patients before and after HCV eradication with DAAs also found that the miRNA profiles were normalized in females after achieving virus eradication, but not completely in males [102]. This study speculated on the possibility that the lack of normalization in males is related to a high risk of developing liver-related complications. We recently determined the levels of three plasma-circulating microRNAs, miRNA-100-5p, miRNA-122-5p, and miRNA-192-5p, in 81 HIV-1/HCV-coinfected individuals on HCV DAA therapy. These three miRNAs were previously shown to highly correlate with the progression of liver fibrosis in PLWH coinfected with HCV [88]. We aimed to explore whether circulating levels of miRNAs could be linked to liver disease progression in HIV-1/HCV-coinfected individuals who eradicate HCV. As expected, HCV eradication was significantly associated with a reduction of the number of circulating miRNA-100-5p, miRNA-122-5p, and miRNA-192-5p in the overall cohort and in individuals with advanced liver fibrosis (unpublished results). Remarkably, no significant reduction in miRNA levels was observed in individuals who did not eradicate HCV. These results indicate that the DAA treatment is linked to a significant reduction in circulating levels of liver disease-associated miRNAs towards the levels seen in healthy donors, and emphasize the utility of circulating miRNAs as biomarkers of liver injury progression.

## 5. Conclusions

New knowledge regarding the role of circulating miRNAs in viral infection and disease is creating opportunities to diagnosis difficult and multifactorial diseases, such as liver disease in PLWH. Although the standardization and validation of circulating miRNA signatures as biomarkers for noninvasive diagnosis of liver injury in PLWH requires further exploration, miRNAs could undoubtedly help provide an improved strategy to identify individuals that may potentially develop harmful liver complications. The circulating miRNA signatures should help in the development of patient risk profiles in a cost-effective and noninvasive manner. miRNAs may increase therapeutic effectiveness and the quality of life of PLWH patients.

## Figures and Tables

**Figure 1 viruses-14-01118-f001:**
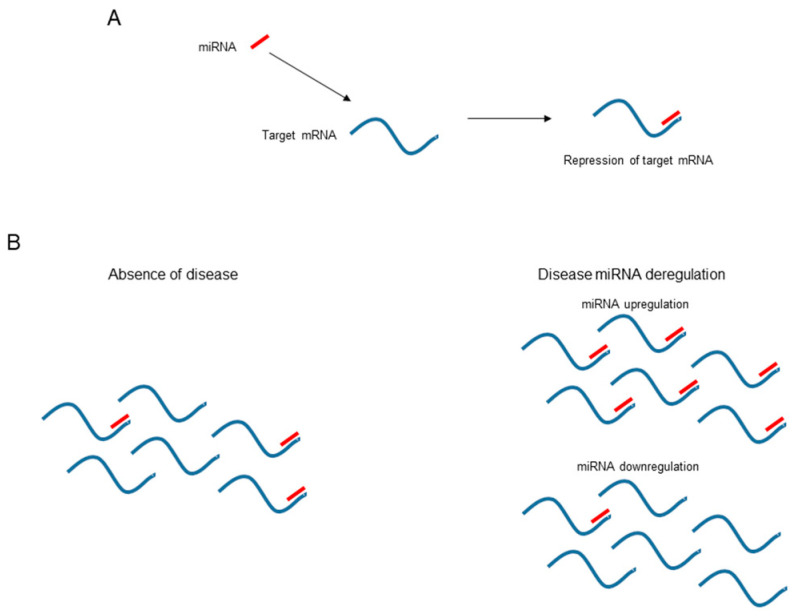
**Molecular mechanism of the repressor activity of microRNAs.** (**A**) miRNAs bind target mRNAs and repress their expression. (**B**) The stoichiometry of target mRNAs and miRNAs governs mRNA expression. In the presence of a disease state, the expression of a miRNA may be upregulated or downregulated and, consequently, the miRNA and target stoichiometry may change and the corresponding mRNA expression be altered. Although not shown in this figure, the opposite situation can also occur; disease may alter the expression of an mRNA and modify the amount of free miRNA, which may have an effect elsewhere.

**Figure 2 viruses-14-01118-f002:**
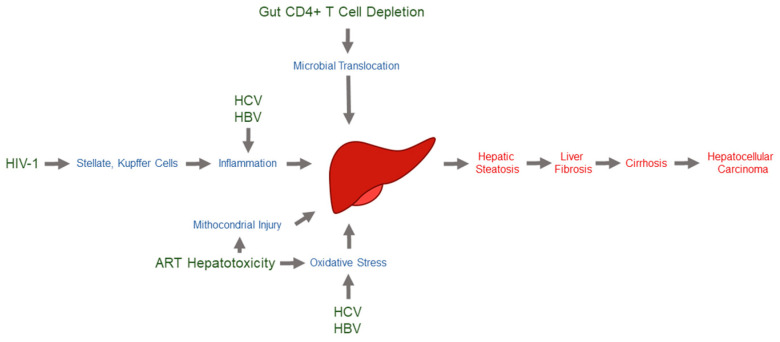
**Most relevant mechanisms of liver injury in people living with HIV-1.** Mechanisms by which different disease entities may cause hepatic injury and fibrosis include oxidative stress, mitochondrial injury, lipotoxicity, immune-mediated injury, cytotoxicity, toxic metabolite accumulation, gut microbial translocation, systemic inflammation, senescence, and nodular regenerative hyperplasia. HIV-1 may use any number of these mechanisms to exert an effect on the liver [19].

**Figure 3 viruses-14-01118-f003:**
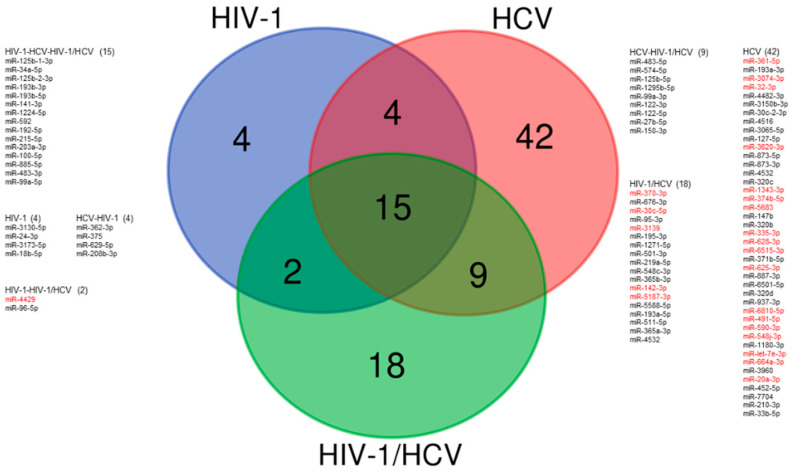
**Signatures of deregulated circulating microRNAs in different infected populations.** Venn diagram showing circulating plasma miRNAs with significant (fold change > 2 and adjusted *p* < 0.05) differences between healthy samples (*n* = 21) and different infected populations: HIV-1-monoinfected (*n* = 53), HCV-monoinfected (*n* = 17), and HIV-1/HCV-coinfected (*n* = 50). Data were obtained from two studies [65,88]. miRNA profile data were generated by small RNA sequencing of individual plasma samples. Downregulated circulating miRNAs are shown in red and upregulated circulating miRNAs in black.

**Table 1 viruses-14-01118-t001:** Most relevant microRNAs implicated in liver metabolism and deregulated in liver disease.

MicroRNA	Target	Liver Expression	Circulating Level	Reference (PMID)
miRNA-122	Glycogen metabolism Glycolysis Lipogenesis Endoplasmic reticulum stress β-oxidation Lipid uptake	Downregulated	Upregulated	16459310
				16258535
				28802563
				19030170
				24973316
				21886843
				27956809
				26565986
				29848284
				30779441
miRNA-34a	Endoplasmic reticulum stress Lipid export β-oxidation	Upregulated	Upregulated	28167956
				19030170
				30142428
				21886843
				23727030
				27956809
miRNA-33	Glucogenesis Lipogenesis Lipid export β-oxidation	Upregulated	Upregulated	24100264
				27669236
miRNA-21	Lipogenesis β-oxidation Lipid uptake	Upregulated	Upregulated	21636785
				19030170
				26338827
				35157721
				23727030
miRNA-375	Autophagy	Downregulated	Upregulated	22504094
				19030170
				30142428
				26874844
				24973316
miRNA-192	Glycolysis Lipogenesis	Upregulated	Upregulated	24973316
				28483554
				30142428
				24973316
				30779441
				27956809
				26565986
miRNA-100	Glycolysis Lipogenesis	Upregulated	Upregulated	24244722
				19030170
				25519019
				29807039
miRNA-155	Glycolysis Lipogenesis	Upregulated	Upregulated	23991091
				28942920
				26948494
				26867493
				35157721
				33569439
miRNA-29	Glycolysis Lipogenesis	Downregulated	Downregulated	28664184
				19372595
				35157721
				29848284
miRNA-223	Inflammation	Upregulated	Downregulated	30964207
				19030170
				32330203

**Table 2 viruses-14-01118-t002:** Most relevant deregulated circulating microRNAs associated with liver injury in people living with HIV-1.

	MicroRNA	Reference (PMID)
**HIV-1 monoinfection**	miRNA-99a-5p, miRNA-100-5p, miRNA-122-3p, miR-125b-1-3p, miRNA-192-5p, miRNA-193b-5p, miRNA-200a	29807039, 29404431, 28883647
**HIV-1/HCV coinfection**	miRNA-22, miRNA-34a, miRNA-99a-5p, miRNA-100-5p, miRNA-122–5p, miRNA-122, miRNA-125b-2–3p, miRNA-192–5p, miRNA-194–5p, miRNA-1246	25125218, 33813557

## Data Availability

All data will available after request.

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
