# Peer review of "Circulating MicroRNAs as a Tool for Diagnosis of Liver Disease Progression in People Living with HIV-1"

_viruses, 2022, doi:10.3390/v14061118_

Round 1

Reviewer 1 Report

The review deals with the usefulness of circulating microRNA markers for the early diagnosis of liver disease in people living with HIV-1. The short review describes the molecular mechanism of microRNA regulatory activity, and liver damage with HIV-1 monoinfected and/or HCV, HBV co-infected individuals. Also, evaluates recent literature on the role of microRNAs as biomarkers of liver disease progression. It describes the deregulated circulating microRNA expression in liver diseases.

Comments:

-Table 1 that shows the most relevant microRNAs implicated in liver metabolism and deregulated in liver disease must be formatted.

-Regarding the progression of liver disease as severity of mild fibrosis and cirrhosis the authors should mention a work ((doi.org/10.1016/j.bbrep.2020.100814) that corroborates the dysregulation four upregulated microRNAs (34a-5p; 885-5p; 193b-3p; 215-5p) found in patients coinfected with HIV-1-HCV-HIV-11/HCV in studies by the authors.The work by Cabral et al., identified 163 mature miRNAs in the mild/moderate fibrosis group and 171 in the cirrhosis group, with 144 in common to both groups of monoinfected HCV patients. Differential expression analysis revealed 5 upregulated miRNAs and 2 downregulated miRNAs in the cirrhosis group relative to the mild/moderate fibrosis group. Differentially expressed circulating miRNAs (upregulated: miR-215-5p, miR-483-5p, miR-193b-3p, miR-34a-5p, miR-885-5p, and downregulated: miR-26b-5p and miR -197- 3p).

-Table 2 and Figure 3 are based on two previous studies by the authors. The circulating plasma miRNAs differences between healthy samples (n=21) and different infected populations: HIV-1-monoinfected (n=53), HCV-monoinfected (n =17), and HIV-1/HCV-coinfected (n=50). The legend of the intersections of the Venn diagram of HIV-1-monoinfected/HCV-monoinfected; HIV-1-monoinfected/HIV-1/HCV-coinfected; HIV-1/HCV-coinfected/HCV-monoinfected and so on, is confusing for the reader due to the combination of the “-“ and “/”.  Please, rename them.  

-Venn diagram: The HCV-monoinfected individuals presents a higher number of deregulated microRNA (70) than HIV-1-monoinfected (25) and HIV-1/HCV-coinfected patients (44). Is there some explanation for this? Also, are HIV-1-monoinfected and HIV-1/HCV-coinfected downregulated microRNAs rare compared with healthy individuals? The authors should address these results as a discussion/conclusion.

Reviewer 2 Report

The authors have made significant contributions in identifying signature patterns of micro RNAs in viral infections. For example the severity of liver damage in the case of HIV co-infected hepatitis C patients and also in determining response to antiviral treatment. Here they present an exhaustive review of micro RNAs involved in HIV-HCV patients and patients with other liver diseases. There is a great interest in clinical practice. The patterns of circulating microRNAs may be an important non-invasive method for evaluating the outcome of liver disease in HIV patients. This may be especially important in numerical terms in HIV patients with non-viral liver pathology, non-alcoholic fatty liver disease, etc. Independently, the identification of these patterns opens a way of interpreting intercellular communication - as if they were hormones - in the terms used by the authors. But in differing to classical hormones, the reception of microRNAs is through W:C base pairs in mRNA substrates. Meaning this type of communication is digital, unlike hormones, which is analog. The advantage n is that it is susceptible to advances in computation that may in the future translate the patterns of microRNAs into patterns of changes in cellular and tissue mRNA expression. I think that apart from clinics, this is a new entry point into the semiotics of biology. For example, how different tissues respond to changes in a microRNA pattern. That is, what is the role of context in signal interpretation. This is just a comment. The field of study is fascinating.